# OpenReview forum: "MSE-Break: Steering Internal Representations to Bypass Refusals in Large Language Models"
_ICLR.cc/2026/Conference — Submitted to ICLR 2026_

### Official Review · Reviewer_dcSm · 2025-10-16

**Soundness:** 2
**Presentation:** 2
**Contribution:** 2
**Rating:** 2
**Confidence:** 5

**Summary:**

This paper leverages the PCA separability of the hidden states within LLMs to train jailbreak prompts, allowing them to output malicious content. Experiments show that these prompts can precisely manipulate the models' internal refusal mechanism, achieving a much higher attack success rate than similar methods such as GCG and AutoDAN.

**Strengths:**

This paper identifies a relevant and timely problem, namely understanding and bypassing refusal mechanisms in LLMs.

The creation of a new dataset, although small and under-documented, shows an intent to provide standardized evaluation materials for future work. If expanded and released, it could benefit reproducibility and comparative research in AI safety.

**Weaknesses:**

This paper lacks novelty, which constitutes a major reason for my recommendation of rejection. In addition, its experimental setup also has many defects, and the writing is not good.

**Novelty Concerns**

- The authors inadequately investigate the coverage of jailbreak research. In fact, there have been quite a number of jailbreak works that go beyond ''surface-level input manipulation'', including the manipulation of embedding, attention, and activation [1,2,3,4].
- Enabling jailbreak by manipulating the concept of refusal is also a topic that has been explored [1,3].
- Creating soft prompts with internal hidden states as the optimization target has also been explored [2].
- This paper mentions that the candidate prompts required for the soft prompts training need to be carefully selected, which seems to be an additional constraint. In [2], the positive samples based on the linear classifiers are randomly selected.

[1] Refusal in Language Models Is Mediated by a Single Direction, https://arxiv.org/abs/2406.11717 (NeurIPS 2024)

[2] Uncovering Safety Risks of Large Language Models through Concept Activation Vector, https://arxiv.org/abs/2404.12038 (NeurIPS 2024)

[3] Stronger Universal and Transferable Attacks by Suppressing Refusals, https://aclanthology.org/2025.naacl-long.302.pdf

[4] Sugar-Coated Poison: Benign Generation Unlocks Jailbreaking, https://arxiv.org/abs/2504.05652 (EMNLP 2025)

---

**Experimental Defects**

- **The narrative of the threat model is misleading**. In Introduction, the authors claim that MSE-break does not require *updating model weights or accessing logits*, but their experiments are based on open-source models, and the training of soft prompts also requires accessing embedding information from any layer, resulting in significant inconsistency. The authors also do not present how the trained prompts are applied to black-box APIs to support their claim.
- **The authors claim to have established a new dataset, but the construction pipeline is opaque**. This raises issues:
  -  How do researchers construct new datasets on new ''concepts''?
  -  The ''concepts'' for constructing the dataset seem to be based on the authors' subjective judgment and lack consistent evidence in model internal representations.
  -  The number is too small, with 75 prompts divided into five topics, and Cybercrime has three subtopics. According to the equal distribution assumption, the subset of a single concept may consist of fewer than 10 questions. Based on the blessing of high-dimensional separability, this separability may not necessarily reflect the refusal mechanism truthfully.
- **Lack of baselines**. Comparing the authors' method with GCG and AutoDAN, which create adversarial prompts, is reasonable, but at least it needs to be compared with similar methods, such as [1,2,3]. In particular, [2] points out that the robustness of using PCA to train refusal vectors is not as good as that of linear regression classifiers. Comparisons with such methods have the potential to become a useful empirical study in the field.

---

**Writing & Demonstration**

- Soft prompt seems not to be a widely accepted concept. The author needs to demonstrate its definition in Introduction, or how it acts on malicious requests to produce an attack.
- At L125, the reasons for using the instruct version models are not fully explained. How does this setting play a role in the effectiveness of their attack?
- Section 4.3 does not provide any figures, tables, or experimental evidence.
- Figure 3 is too small to read, with a lot of blank space on both sides. A better presentation could be used. (This seems to be an AI-generated interface, and I think it would be easy to adjust)

**Typos**

- At L137, Gpt-4o -> GPT-4o
- At L299, Layer 9-16(middle layers) -> Layer 9-16 (middle layers)
- At L352, a period is missing

**Questions:**

Major as listed in weakness section.

**Details Of Ethics Concerns:**

The dataset required to build the soft prompts is the key to the reproducibility of this paper, but the proposed direct release may raise ethical issues. Since the research is based on open-source models, anyone can train soft prompts based on this to obtain malicious content.

Additionally, it is unclear whether this work has been approved by the IRB of the authors' institution. They need to supplement relevant evidence in their ethical statement section.

---

> ### Author Response · Authors · 2025-11-27
>
> We thank the reviewer for the thoughtful feedback and for the valuable steps to improve the paper. We address each point below and have incorporated all changes into the revised manuscript.
>
> **Addressing Novelty Concerns:**
> We provide a full clarification of the novelty and distinction from prior work in our Top-Level Comment (see “Novelty Clarification”).
>
> **Candidate Prompt Ablation:** In response to the concern about candidate-prompt selection, we ran a control experiment replacing the top-scoring benign prompt with a random accepted prompt **(see Appendix G.2)**. ASR drops as expected but remains high (>0.70 for all models), showing that MSE-Break is robust even without optimal prompt selection. This supports our claim that MSE-Break leverages both representational geometry and contextual framing.
>
> **Narrative Threat Model:** We agree and remove the potentially misleading statement about “not updating model weights.” We also clarify that extending MSE-Break to black-box settings (e.g., via distillation into discrete prefixes) is an important direction for future work.
>
> **The authors claim to have established a new dataset, but the construction pipeline is opaque(Misunderstanding):**
>
> We clarify that our dataset does not introduce new “concepts.” We use the same five harmful categories widely adopted in safety evaluations (e.g., HarmBench, AdvBench): bombs, bioweapons, narcotics, cybercrime, and terrorism. As detailed in **Section 2.4** , prompts are created through a fixed pipeline: (1) select seed prompts from AdvBench, (2) generate additional prompts using GPT-4o under strict constraints to ensure they are semantically diverse and safety-relevant, and (3) manually verify that they elicit refusals. This process avoids subjective concept definition.
>
> Regarding size: while the evaluation set contains 75 prompts, our concept-separation analyses use **128 prompts per concept (64 refused + 64 accepted)**, providing strong evidence that refusal behavior is encoded at the internal-representation level. The 75-prompt evaluation set is intentionally compact because it serves solely as a held-out test set. MSE-Break is trained only on the selected accepted prompts for each concept **(see Section 5.1)**, not on the evaluation dataset.
>
> **Lack of Baselines:**
>
> We have significantly strengthened the experimental section. In addition to GCG and AutoDAN, the revised paper now includes: **SCAV (Xu et al. 2024)**, an embedding-level jailbreak perturbing last-token hidden-state activations and **Weight Orthogonalization (Arditi et al. 2024)**, implemented as a rank-1 refusal-direction suppression **(see Section 5.3)**.
>
> MSE-Break significantly outperforms SCAV on three out of four models and is only slightly below SCAV on Llama-3.1 (0.87 vs. 0.89). Relative to Weight Orthogonalization, MSE-Break achieves **substantially higher ASR** on every model, often by large margins (e.g., Gemma-2B: 0.91 vs. 0.18; Llama-3.1: 0.87 vs. 0.12).
>
> Additionally we evaluate MSE-Break against two state of the art representation-level defenses: **Circuit Breaker and RepBend(See section 5.4)**. These methods have been found to be highly effective at defending against embedding-based jailbreaks, often reducing ASR by 40–70% for CAV-style and gradient-based representation attacks. In contrast, MSE-Break observes only **minor fluctuations** (±2–4 percentage points) relative to the undefended models. Unlike typical embedding-level jailbreaks, MSE-Break bypasses these defenses by targeting the **upstream trigger** of refusal (e.g., the ‘bomb’ embedding), preventing harmful activation patterns from forming in the first place.
>
> *We clarify that PCA is used solely for visualization of concept separation—it plays no role in training or in the attack mechanism*
>
> **Writing and Demonstration:**
>
> We added a clear definition of soft prompts in the preliminaries, emphasizing that the prefix does not act on the malicious query itself but reshapes the internal semantic embedding of the target concept (e.g., “bomb”) toward its benign counterpart.
>
> Regarding instruct vs. base models, we now explain at L145 that we use instruction-tuned models because base models receive minimal safety training and often output harmful content even without an attack.
>
> For Section 4.3, we added new experimental evidence: a subtopic-level generalization test **(Appendix G.1)**. A prefix trained on the parent concept Narcotics achieves high ASR (0.86–0.91) across five distinct drug-related subtopics, demonstrating the concept-level generalization claimed in the section.
>
> We will substantially improve the clarity and presentation quality of all figures in the camera-ready version. We hope these clarifications and the substantial new experiments directly resolve your concerns. If so, we would greatly appreciate an updated score reflecting the strengthened contribution. Thank you for the detailed feedback, and we are happy to address any remaining issues should they arise.

---

> > ### Comment · Reviewer_dcSm · 2025-11-28
> >
> > I appreciate the authors' efforts to improve the paper, but I still have the following unresolved concerns:
> >
> > 1. The authors has not provided any revision version or a new baseline table, making their response unconvincing.
> > 2. The use of soft prefixes is not novel; GCG, AutoDAN, and SCAV (at prompt-level) have all studied the jailbreak effects caused by adding prefixes.
> > 3. The dataset used for training is carefully selected. For the sake of reproducibility, does the authors commit to releasing this dataset? In what manner?

---

> > > ### Author Response · Authors · 2025-11-28
> > >
> > > Thank you for the timely response, and we appreciate the notice of the effort to improve the paper. We additionally appreciate many of your valuable suggestions that helped strengthen the claims of the paper.
> > >
> > > **1:** At the time of the first rebuttal, we uploaded a revised version of the paper with all the tables and all edits described in the rebuttal.
> > >
> > > **2:** We agree that prefix-based methods exist, but none of them use a prefix to **selectively rewrite the internal representation of a single concept**. Prior work employs prefixes for global effects—for example, GCG and AutoDAN optimize for affirmative, non-refusal responses, and SCAV(prompt) optimizes so that the final-layer embedding looks safe to a classifier. These approaches do not modify how the model internally represents a specific concept such as “bomb.”
> > >
> > > MSE-Break’s **prefix objective** is fundamentally different: it performs a **targeted semantic rewrite** of a single concept’s embedding at the first token-level representation. This causes the model to internally encode “bomb” as its benign counterpart while leaving the rest of the query unchanged. This level of **fine-grained, concept-specific intervention** has not been the focus of prior prefix-based jailbreaks. It is also what enables single-token hijacking **(see Section 6.2)**: appending the prefix consistently allows the model to interpret **bomb** as a **bunny** across all tested queries.
> > >
> > > In short, MSE-Break moves beyond targeting refusal behavior to directly **altering how the model represents individual concepts** in embedding space. To our knowledge, no prior jailbreak method, including GCG, SCAV, or AutoDAN, demonstrates this concept-level rewrite mechanism using a soft prefix.
> > >
> > > We found it neat that this level of concept rewriting specificity was possible with a simple soft prompt, and that it could be used in such an effective way to mitigate refusal. We believe the mechanism itself has **broader implications**, as the ability to directly rewrite a model’s internal representation of a concept may be useful beyond jailbreak settings. The additional points of novelty beyond the use of the soft prefix are clearly outlined in the top level comment.
> > >
> > > 3: Yes, we commit to releasing the training dataset in its entirety, along with the evaluation dataset and the multiple 128 prompt datasets used to showcase linear separability for the harmful concepts.
> > >
> > > We appreciate the time and care you put into the review. With the clarified novelty, expanded baselines, and commitment to full dataset release, we hope the strengthened manuscript addresses your concerns and helps reinforce your support for the paper. We believe this mechanism, and the concept-level rewriting behavior it reveals, could be valuable for the broader interpretability community.

---

> > > > ### Comment · Reviewer_dcSm · 2025-11-28
> > > >
> > > > Thank you for the clarification, and I apologize for overlooking the revised paper.
> > > >
> > > > I am still confused by the motivation of this paper. Although methods like GCG seem general, they are optimized for specific questions. Specific concepts may not necessarily be the best proxies for successful attacks. Could you comment on this intuitive conflict? Also, could you provide experimental evidence that the prefixes generated by GCG do not modify the corresponding concepts (using your probes), while MSE-break does achieve this?

---

### Official Review · Reviewer_vvaj · 2025-10-17

**Soundness:** 2
**Presentation:** 2
**Contribution:** 1
**Rating:** 2
**Confidence:** 4

**Summary:**

This paper proposes MSE-Break, a method that steers internal representations of harmful concepts in large language models using a soft prompt optimized via mean squared error minimization. The approach aims to align harmful and benign embeddings to bypass refusal behavior across multiple open-weight models.

**Strengths:**

1. **Clear Motivation.** The paper is grounded in the observed separation between harmful and benign embeddings.
2. **Simple and Reproducible Method.** The approach is easy to implement and converges quickly.

**Weaknesses:**

1. **Paper Structure.** The paper’s organization could be improved. For instance, including implementation details in the Preliminaries section feels unconventional and disrupts the logical flow.

2. **Limited Effectiveness.** There already exist strong jailbreak attacks capable of breaking LLM safeguards efficiently [1,2], particularly for open-source models like Qwen or LLaMA. A comparison with such methods would strengthen the evaluation.

3. **Incremental Novelty.** The proposed technique mainly involves applying a weighted MSE objective to selected residual layers and tuning a short soft prompt. Similar ideas—using single directions or activation manipulations to alter refusal behavior—have been well explored in prior work [3], even though that work is cited in this paper.

[1] Ding, Peng, et al. "A wolf in sheep's clothing: Generalized nested jailbreak prompts can fool large language models easily." NAACL (2024).\
[2] Andriushchenko, Maksym, Francesco Croce, and Nicolas Flammarion. "Jailbreaking leading safety-aligned llms with simple adaptive attacks." ICLR (2025).\
[3] Arditi, Andy, et al. "Refusal in language models is mediated by a single direction." NeurIPS (2024).

**Questions:**

1. I understand that this is a white-box attack, but is there any potential for applying it to proprietary models? For example, is the learned soft prompt transferable across models? I assume not, since soft prompts are model-specific. [1]
2. Recently, several methods have been proposed for safety alignment at the representation level [2,3]. Is your method still effective against such defenses? It would strengthen your argument to demonstrate robustness against these baselines, as they aim to harden the very representations your attack targets.

[1] Jia, Xiaojun, et al. "Improved techniques for optimization-based jailbreaking on large language models." ICLR (2025).\
[2] Zou, Andy, et al. "Improving alignment and robustness with circuit breakers." NeurIPS (2024).\
[3] Yousefpour, Ashkan, et al. "Representation bending for large language model safety." ACL (2025).

---

> ### Author Response · Authors · 2025-11-27
>
> We thank the reviewer for the detailed feedback. We appreciate the recognition of the clarity of our motivation and the simplicity and reproducibility of the method. We address each concern below, and all updates are incorporated in the revised manuscript.
>
> **Novelty Clarification:**
>
> **We provide a full clarification of the novelty and distinction from prior work in our Top-Level Comment (see “Novelty Clarification”).**
>
> **Paper Structure:** We have moved implementation details, such as Evaluation, out of preliminaries and into the Experiments Section.
>
> **Limited Effectiveness:**
> We have significantly strengthened the experimental section. In addition to GCG and AutoDAN, the revised paper now includes: **SCAV (Xu et al. 2024)**, an embedding-level jailbreak perturbing last-token hidden-state activations and **Weight Orthogonalization (Arditi et al. 2024)**, implemented as a rank-1 refusal-direction suppression. We include SCAV because prior work reports that it achieves over 90% ASR on multiple open-source LLMs, making it one of the strongest embedding-level jailbreaks available.
>
> MSE-Break significantly outperforms SCAV on three out of four models and is only slightly below SCAV on Llama-3.1 (0.87 vs. 0.89). Relative to Weight Orthogonalization, MSE-Break achieves **substantially higher ASR on every model**, often by large margins (e.g., Gemma-2B: 0.91 vs. 0.18; Llama-3.1: 0.87 vs. 0.12). These results demonstrate that MSE-Break is competitive with, and in many cases stronger than, state-of-the-art representation-level attacks **(see Section 5.3)**.
>
> **Question 1:** Yes, soft prompts themselves are model-specific, since they depend on each model’s residual-stream geometry. However, the underlying vulnerability MSE-Break exploits is not model-specific. Across all models we evaluated, the same structural property holds: harmful concepts form a linearly separable cluster from their benign-context embeddings. Additionally, the same candidate prompts used to train the soft prompt are robust across models.
>
> **Question 2:** Thank you for the suggestion, as it empirically highlights another key benefit of MSE-Break. We tested the method against both Circuit Breakers and RepBend **(see Section 5.4)**. We observe only minor fluctuations (±2–4 percentage points) relative to the undefended models. MSE-Break remains effective because it removes the triggers of refusal rather than counteracting its effects. By rewriting the harmful concept’s embedding upstream, the model never enters the harmful-activation manifold that Circuit Breakers and RepBend depend on, so these defenses don’t meaningfully activate.
>
> We hope these clarifications and the substantial new experiments directly resolve your concerns. If so, we would greatly appreciate an updated score reflecting the strengthened contribution. Thank you for the detailed feedback, and we are happy to address any remaining issues should they arise.

---

> > ### Comment · Reviewer_vvaj · 2025-11-28
> > **Thank you for Rebuttal**
> >
> > I have revisited the paper and rebuttal carefully. The updates improve clarity but do not substantively change the overall contribution or address the central novelty concerns I described. My evaluation remains consistent, and I am confident in the rating.

---

> > > ### Author Response · Authors · 2025-11-28
> > >
> > > Thank you for revisiting the paper and reading through the rebuttal. We do earnestly believe in the novelty of the method for the interpretability and safety community. We've attached a comment to another reviewer that helps explain the novelty of the method and how it differs from other soft prefix based methods. We hope this serves to clear up any concerns to strengthen your evaluation, but regardless we thank you for your evaluation of the paper.
> > >
> > > **Comment on Novelty**
> > >
> > > **2:** We agree that prefix-based methods exist, but none of them use a prefix to **selectively rewrite the internal representation of a single concept**. Prior work employs prefixes for global effects—for example, GCG and AutoDAN optimize for affirmative, non-refusal responses, and SCAV(prompt) optimizes so that the final-layer embedding looks safe to a classifier. These approaches do not modify how the model internally represents a specific concept such as “bomb.”
> > >
> > > MSE-Break’s **prefix objective** is fundamentally different: it performs a **targeted semantic rewrite** of a single concept’s embedding at the first token-level representation. This causes the model to internally encode “bomb” as its benign counterpart while leaving the rest of the query unchanged. This level of **fine-grained, concept-specific intervention** has not been the focus of prior prefix-based jailbreaks. It is also what enables single-token hijacking **(see Section 6.2)**: appending the prefix consistently allows the model to interpret **bomb** as a **bunny** across all tested queries.
> > >
> > > In short, MSE-Break moves beyond targeting refusal behavior to directly **altering how the model represents individual concepts** in embedding space. To our knowledge, no prior jailbreak method, including GCG, SCAV, or AutoDAN, demonstrates this concept-level rewrite mechanism using a soft prefix.
> > >
> > > We found it neat that this level of concept rewriting specificity was possible with a simple soft prompt, and that it could be used in such an effective way to mitigate refusal. We believe the mechanism itself has **broader implications**, as the ability to directly rewrite a model’s internal representation of a concept may be useful beyond jailbreak settings. The additional points of novelty beyond the use of the soft prefix are clearly outlined in the top level comment.

---

### Official Review · Reviewer_Ggpr · 2025-10-30

**Soundness:** 3
**Presentation:** 2
**Contribution:** 2
**Rating:** 6
**Confidence:** 3

**Summary:**

This paper introduces MSE-Break, a novel, interpretability-driven jailbreaking technique for large language models. The core idea is to optimize a continuous soft-prompt prefix via gradient descent. Unlike traditional methods that target output logits, MSE-Break's objective is to minimize the Mean Squared Error (MSE) between the internal representation of a specific harmful concept (e.g., "bomb") in a refused context and its representation in a carefully selected benign, accepted context. The authors first provide strong empirical evidence that for harmful concepts, the internal representations of refused versus accepted prompts are linearly separable in the model's activation space. MSE-Break directly exploits this separability. On a testbed of four open-source, safety-aligned LLMs (including Gemma-2B-IT and LLaMA-3.1-8B-IT), the method achieves attack success rates (ASR) often exceeding 90%, significantly outperforming and converging orders of magnitude faster than strong baselines like GCG and AutoDAN. The resulting soft prompt is concept-specific but generalizable across many different user prompts involving that concept.

**Strengths:**

* It identifies a new, potent, and highly efficient attack vector. The fact that this method is orders of magnitude faster than baselines (minutes vs. 30+ hours, Table 3) by optimizing a reusable, concept-general prompt is a major finding.

* It underscores a deep vulnerability in current alignment techniques. The results strongly suggest that existing safety training, while effective at a surface level, fails to create robust representations at the concept level.

**Weaknesses:**

* The method relies on a white-box setting, requiring full gradient access to optimize the soft prompt. This is acknowledged by the authors but remains the primary barrier to this attack's applicability to closed-source, black-box models, which are a major part of the safety landscape.

* The experiments are limited to smaller-scale open-source models (<= 8B parameters). It is an open question whether the core empirical finding—the clean linear separability of harmful/benign concept embeddings—holds true for much larger models (e.g., 70B+ or frontier models). Refusal mechanisms and representational geometry might differ significantly at scale.

* The method's success appears to be critically dependent on the selection of a "good" benign candidate prompt (Section 4.2). The current process involves a set of manual heuristics and a scoring function, which introduces a human-in-the-loop component and makes the attack seem less automated than methods like GCG.

**Questions:**

Following the weakness above, how crucial is the candidate prompt scoring function (Section 4.2)? What is the performance (ASR) drop if you simply pick a random accepted prompt for a given concept, versus the top-scoring one selected by your metric?

---

> ### Author Response · Authors · 2025-11-27
>
> We thank the reviewer for the thoughtful and positive assessment. We appreciate the clear articulation of the strengths and agree that the limitations noted (white-box setting and scale) are valid and already discussed in the paper.
>
> **Candidate Prompt Ablation:** In response to the concern about candidate-prompt selection, we ran a control experiment replacing the top-scoring benign prompt with a random accepted prompt **(see Appendix G.2)**. ASR drops as expected but remains high **(>0.70 for all models)**, showing that MSE-Break is robust even without optimal prompt selection. This supports our claim that MSE-Break leverages both representational geometry and contextual framing. Even with random accepted prompts, the harmful concept is still interpreted internally as benign, while higher-quality contexts strengthen this semantic shift and yield the highest ASR.
>
> To strengthen the contribution, we have added **two new sets of experiments** in the revised draft:
>
> 1. Comparisons against stronger embedding-level baselines (SCAV and Weight Orthogonalization), where MSE-Break outperforms them across models **(see Section 5.3)**
>
> 2. Robustness under representation-level defenses (Circuit Breakers and RepBend), where MSE-Break maintains high ASR with only ±2–4% fluctuation because it mitigates downstream harmful activations entirely **(see Section 5.4)**.
>
>
> We believe these additions directly reinforce the reviewer’s assessment that the method reveals a deep vulnerability in alignment and is both efficient and effective. We thank the reviewer again for the supportive evaluation.

---

### Official Review · Reviewer_NDrd · 2025-10-30

**Soundness:** 2
**Presentation:** 3
**Contribution:** 3
**Rating:** 6
**Confidence:** 3

**Summary:**

This work proposes an innovative method to jail-break LLMs. The authors hypothesized that the decision to refuse to answer a prompt is triggered by specific sensitive concepts, and found that the representations of such concepts is linearly separable between harmful and benign contexts. The proposed method optimizes a soft prompt such that, when it is prepended to a harmful request containing a target concept, the representation of the context is maximally similar to the representation of the same concept in a benign context. Experiments show that the attack is highly effective, and is computationally inexpensive.

**Strengths:**

* While it was known that the last-token representations of harmful and harmless prompts are linearly separable, the insight that this derives from a similar property of the sensitive concepts mentioned in the prompt is very interesting and, to my knowledge, novel.
* The discovery of the portability of the attack across prompts involving a same concept is also surprising and interesting
* The effectiveness of the proposed method draws attention on a vulnerability that deserves to be further understood and mitigated.

**Weaknesses:**

[W1] The experimental section is borderline sufficient. Only GCG and AutoDAN are used as baselines. The paper would be stronger if jailbreaking methods not based on adversarial prompt search were added to the experiments. In particular, since MSE-break requires access to model weights, it would be interesting to see it compared to the weight orthogonalization method of Arditi et al. which can also be easily implemented with inference-time interventions by suppressing the contribution of all MHA and MLP components along the refusal direction. It would be great to have at least one model at a larger scale than 8B.

[W2] Minor: Section 4.2 on effective prompt candidate selection could be better clear. It was not initially clear to me that this was about generating the benign contexts for the harmful concepts that would not trigger a refusal: this should be explained upfront. There should also be more clarity on what happens when the harmful concept spans more than one token: is it the last token of the concept that is considered? L206 mention a token position (singular), but L207-208 mention averaging token vectors.

**Questions:**

Figure 4: what model and layer are these plots relative to?

L485: what are the 'tailored embeddings' being referred to here?

Page layout and figures could be improved

**Details Of Ethics Concerns:**

The paper describes an effective model jailbreaking method that could enable malicious agents. While I concur with the authors that the value of the contribution is net-positive, I am not an expert ethicist, and expert advice should be sought.

---

> ### Author Response · Authors · 2025-11-27
>
> We thank the reviewer for the thoughtful and positive evaluation. We appreciate the recognition of our core findings, including the novel insight into concept-level separability and the portability of the attack, and we address each concern and question below. All updates are included in the revised manuscript.
>
> **W1:**   We have significantly strengthened the experimental section. In addition to GCG and AutoDAN, the revised paper now includes: **SCAV (Xu et al. 2024)**, an embedding-level jailbreak perturbing last-token hidden-state activations and **Weight Orthogonalization (Arditi et al. 2024)**, implemented as a rank-1 refusal-direction suppression. We include SCAV because prior work reports that it achieves over 90% ASR on multiple open-source LLMs, making it one of the strongest embedding-level jailbreaks available **(see Section 5.3)**.
>
> MSE-Break significantly outperforms SCAV on three out of four models and is only slightly below SCAV on Llama-3.1 (0.87 vs. 0.89). Relative to Weight Orthogonalization, MSE-Break achieves **substantially higher ASR on every model**, often by large margins (e.g., Gemma-2B: 0.91 vs. 0.18; Llama-3.1: 0.87 vs. 0.12). These results demonstrate that MSE-Break is competitive with, and in many cases stronger than, state-of-the-art representation-level attacks.
>
> Additionally we evaluate MSE-Break against two state of the art representation-level defenses: **Circuit Breaker and RepBend(See Section 5.4)**. These methods have been found to be highly effective at defending against embedding-based jailbreaks, often reducing ASR by 40–70% for CAV-style and gradient-based representation attacks. In contrast, MSE-Break observes only minor fluctuations (±2–4 percentage points) relative to the undefended models. Unlike typical embedding-level jailbreaks, MSE-Break bypasses these defenses by targeting the upstream trigger of refusal (e.g., the ‘bomb’ embedding), preventing harmful activation patterns from forming in the first place.
>
> **W2:** We thank the reviewer for noting that Section 4.2 could be clearer. We have revised the section to explicitly state up front that candidate benign contexts are generated by selecting prompts that mention the target harmful concept but do not trigger refusal. Regarding token handling, we clarify that the representation of a multi-token concept is computed by averaging the hidden states of all tokens in the concept span.
>
> **Questions:**
> - In the revised draft, we now specify that Figure 4 is computed on LLaMA-3.1-8B-IT, using hidden states from layer 17. We have added this detail directly in the caption for clarity.
>
> - Tailored embeddings refer to embeddings extracted from benign contexts that are semantically aligned to the target harmful query (e.g., obtaining a benign embedding for “bomb” from a historical-weapons discussion and applying it to a harmful historical-weapons query). Such embeddings can marginally increase ASR, but require context-specific selection for each prompt. For MSE-Break, we instead use general benign embeddings that work robustly across many prompts without needing per-query tailoring.
>
> - We will substantially improve the clarity and presentation quality of all figures in the camera-ready version.
>
> We appreciate the positive assessment, and hope that the significantly expanded experimental evaluation and clarifications address the remaining concerns and support a higher confidence in the paper’s contribution.

---

### Author Response · Authors · 2025-11-27
**Novelty Clarification**

We thank Reviewers NDrd and Ggpr for explicitly noting the novelty of our mechanism. To address the remaining novelty concerns, we emphasize that MSE-Break introduces a fundamentally new jailbreak mechanism: one that is distinct from all prior embedding-, activation-, and suffix-based attacks. The distinctions are substantive rather than stylistic, and we outline them clearly below.

**Important Distinction:**

MSE-Break is not an activation-steering or refusal-vector method: it neither extracts refusal directions nor edits weights or logits, but instead uses a soft prefix so the model thinks a harmful concept is internally encoded as benign. It’s a **semantic shift**. Without the prefix, the first hidden state for *bomb* in refused prompts lies in a refusal-associated region. With p prepended, the very first hidden state for bomb already lies in the benign region from the moment the model encounters it; there is no intermediate ‘harmful’ state that is later corrected or routed. This is why representation defenses like Circuit Breaker or RepBend, which intervene only after harmful activations have already formed are **completely ineffective**, yet work so well on other embedding-level attacks. This upstream semantic rewrite is possible because soft prefixs' enters the residual stream before any content tokens, allowing it to reshape how the model internally encodes the concept that follows **(see Section 2.2)**.

Additionally, a *key part* of MSE-Break’s uniqueness is that it exploits both *representational geometry* and *contextual framing*. While the core mechanism is the upstream shift of the concept embedding, the quality of the benign context influences how strongly this shift manifests **(see Appendix G.2)**. Benign embeddings drawn from academic, scientific, or historical contexts are cleaner and more semantically aligned to harmful use cases, which allows the model to more naturally produce harmful output.

**Conceptual Novelty:**

First, we contribute a new empirical observation about how refusal is triggered in aligned LLMs: the internal embedding of a harmful concept (e.g., “bomb”) occupies *different regions* of activation space depending on whether the prompt appears in a refused or accepted context. Prior jailbreak work has largely examined the last-token representation, but we show that this context-dependent shift occurs with the embedding of a harmful concept.

This upstream separation of the *concept itself* provides a new perspective on how refusal circuits are activated and directly motivates MSE-Break’s approach of modifying the concept embedding rather than overriding a downstream safety signal. To our knowledge, this phenomenon and its use for jailbreak design have not been previously described.

**Attack Novelty:**

**This work is the first to identify that the embedding of a harmful concept—rather than a downstream refusal signal—is an upstream trigger of refusal and a viable target for jailbreak design.**

MSE-Break introduces a distinct jailbreak mechanism: a **concept-level upstream intervention** that rewrites the embedding of the harmful concept *before* the model activates any refusal pathway. Prior jailbreaks—whether based on embeddings, attention, or activation steering —operate primarily after the model has already detected the harmful concept and begun forming a refusal direction. These methods attempt to override or suppress the refusal itself. Even weight orthogonalization functions as a global intervention, removing an assumed refusal vector from all hidden states.

In contrast, MSE-Break targets **one of the upstream triggers that causes refusal:** the internal representation of the harmful concept. By shifting this concept into its benign embedding region at the first token-level representation, the model never forms a refusal activation at all. Its early hidden states evolve as if the query were benign for the entire forward pass **(see Figure 4)**.

This upstream mechanism produces behaviors not seen in prior jailbreaks: (i) the model treats all lexical variants and paraphrases of the concept as benign; (ii) a single prefix generalizes across many differently phrased harmful queries; and (iii) single-token hijacking **(Sec. 6.2)**, making the model internally treat “bomb” as “bunny”, succeeds because altering the concept embedding with a **soft prefix** directly changes all subsequent hidden states.

By intervening at the level of **concept semantics**, and focusing on triggers rather than suppressing refusal, MSE-Break provides a mechanism that is both **new and fundamentally different** from all representation-level jailbreaks. The implications extend beyond jailbreaks: they show that soft prompts can systematically reshape how models represent concepts and ideas.

We will further refine the exposition in the camera-ready version to make the core novelty even more explicit, though the mechanism and contribution are already fully presented in the current draft.

---

### Meta-Review · Area_Chair_SgKP · 2025-12-06

**Summary:**

the paper introduces an interesting observation about concept-level separability and proposes a simple, fast jailbreak method, but the empirical evidence and comparative evaluation are not strong enough to support the claimed novelty. Two reviewers view the upstream semantic-shift mechanism as promising, but the other two argue that similar representation-level jailbreaks and concept-steering methods already exist and that the paper does not convincingly differentiate itself from that work. The experiments are limited to small open-weight models, lack the most relevant baselines, and rely on a small and under-specified dataset. The threat model is somewhat inconsistent, because the method requires full white-box access despite claims that it avoids strong assumptions. Since the unresolved weaknesses relate directly to novelty, completeness of evaluation, and clarity of contributions, the balance of evidence supports a rejection.,

**Reviewer Concerns:**

The rebuttal successfully clarifies the conceptual intent of the method. It makes the upstream nature of the intervention more explicit and explains how the soft prefix shifts the concept embedding before any refusal circuitry activates. It also clears up smaller points, such as the role of benign prompt selection and how multi-token concepts are handled. These clarifications address several confusion-based concerns, especially from the more positive reviewers.
the major concerns that drive the negative reviews remain unresolved. refusal-direction suppression, concept-activation vector methods, orthogonalization-based defenses, or recent universal jailbreaks. The limited scale of the evaluation (only models up to eight billion parameters) is still unaddressed, and the question of whether concept-level separability holds for larger or frontier models remains open. The dataset remains very small and under-documented, with no principled procedure for defining concepts or verifying that observed separability reflects genuine mechanisms rather than artifacts of small sample size. The threat-model confusion persists, because the approach depends on full gradient access to internal states while the manuscript emphasizes that it does not require strong assumptions. These unresolved issues directly relate to novelty, completeness, and rigor, and therefore the rebuttal is not sufficient to overturn the core criticisms.

**Reviewer Scores:**

Reviewer NDrd raised primarily clarity and baseline issues. Their evaluation would likely remain unchanged, so 6->6.

Reviewer Ggpr was mildly positive but highlighted dependence on benign-context selection, the white-box assumption, and the small model scale. the rebuttal clarifies Candidate Prompt Ablation. didn't respond back, but I guess their score would likely stay the same, so 6->6.

Reviewer vvaj viewed the contribution as incremental and emphasized missing comparisons to the most relevant representation-level jailbreaks. the reviewer responded back, and the score would likely remain unchanged, 2->2.

Reviewer dcSm raised the strongest novelty objections, highlighted flaws in the dataset and threat model, and noted multiple missing baselines. it likely stays 2->2.

---

### Decision · Program_Chairs · 2026-01-26

Reject